# Let LLM Tell What to Prune and How Much to Prune

**Mingzhe Yang** [* 1]   **Sihao Lin** [* 2]   **Changlin Li** [3]   **Xiaojun Chang** [1]

## Abstract

Large language models (LLMs) have revolutionized various AI applications. However, their billions of parameters pose significant challenges for practical deployment. Structured pruning is a hardware-friendly compression technique and receives widespread attention. Nonetheless, existing literature typically targets a single structure of LLMs. We observe that the structure units of LLMs differ in terms of inference cost and functionality. Therefore, pruning a single structure unit in isolation often results in an imbalance between performance and efficiency. In addition, previous works mainly employ a prescribed pruning ratio. Since the significance of LLM modules may vary, it is ideal to distribute the pruning load to a specific structure unit according to its role within LLMs. To address the two issues, we propose a pruning method that targets multiple LLM modules with dynamic pruning ratios. Specifically, we find the intrinsic properties of LLMs can guide us to determine the importance of each module and thus distribute the pruning load on demand, i.e., what to prune and how much to prune. This is achieved by quantifying the complex interactions within LLMs. Extensive experiments on multiple benchmarks and LLM variants demonstrate that our method effectively balances the trade-off between efficiency and performance.

## 1. Introduction

Large language models (LLMs) have demonstrated exceptional performance in the field of natural language processing (Mann et al., 2020; Thoppilan et al., 2022; Zhu et al., 2024). Nonetheless, the billions of parameters (Zhao et al., 2023) characteristic of LLMs substantially increase storage requirements and computational overhead. Therefore, de-

veloping effective methods to compress LLMs has become a valuable research direction.

Network Pruning (Han et al., 2015) is a promising method to reduce model complexity by pruning the non-essential network connections. Traditional pruning strategy (i.e., weight magnitude) may not be suitable to LLMs (Yin et al., 2024; Muralidharan et al., 2024), which typically requires fine-tuning (Gordon et al., 2020; Jaiswal et al., 2023; Xia et al., 2023) to restore the performance. Since retraining LLMs could be prohibitively expensive, (Frantar & Alistarh, 2023) propose SparseGPT, the first accurate one-shot unstructured pruning method without any post-training. Compared to unstructured pruning, ***structured*** pruning (An et al., 2024; Song et al., 2024; Ashkboos et al., 2024; Zhong et al., 2024; Ma et al., 2023; Men et al., 2024) is more friendly for deployment, which establishes a criterion to determine the importance of specific structures, e.g., blocks and layers, and remove the uninformative components accordingly.

Nonetheless, existing literature on structured pruning only focuses on a single structure unit in LLMs, such as transformer block (Song et al., 2024; Men et al., 2024), attention & MLP layers (Zhong et al., 2024) and the rows & columns of weight matrices (An et al., 2024;

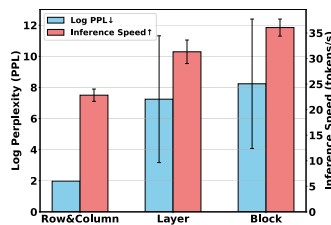

*Figure 1.* Dilemma between performance and efficiency. A faster pruning strategy (speed ↑) comes with inferior performance (PPL ↓).

Ashkboos et al., 2024; Ma et al., 2023; Ling et al., 2024). Different structure units vary from parameter counts and functionality, contributing differently to inference speed versus performance. An example is illustrated in Figure 1 where three structure units, accounting for 10% parameters, of LLaMa2-7B are randomly pruned. The result shows that there is no free lunch for structured pruning when targeting a certain LLM module. Specifically, a faster strategy (e.g., block-wise pruning) comes with worse performance and vice versa. Consequently, previous methods could lead to an imbalance between throughput versus performance, as shown in Figure 2.

In addition, a majority of arts (Ashkboos et al., 2024; Song et al., 2024; Zhong et al., 2024; Lin et al., 2024; Men et al.,

---
[*]Equal contribution [1]University of Science and Technology of China [2]RMIT University [3]Stanford University. Correspondence to: Xiaojun Chang <xjchang@ustc.edu.cn>.

*Proceedings of the 42nd International Conference on Machine Learning*, Vancouver, Canada. PMLR 267, 2025. Copyright 2025 by the author(s).

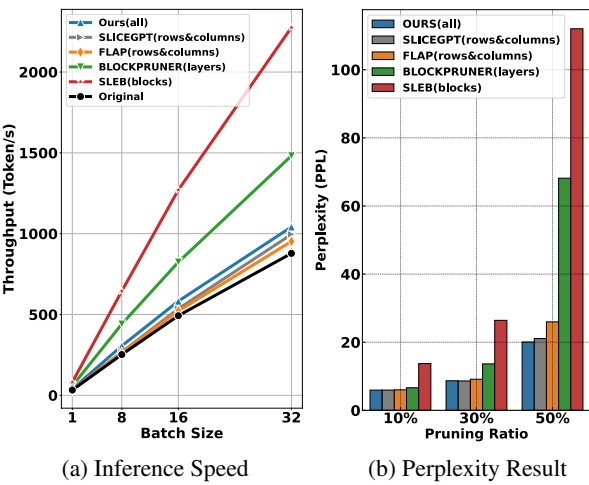

*Figure 2.* Compare different structured pruning methods. (a) We compare the inference speed of pruned model using different structured pruning methods. (b) We evaluate the perplexity performance of different structured pruning methods.

2024) prune LLMs with a prescribed ratio in each unit. For instance, SLEB (Song et al., 2024) uses a binary operator to determine whether to prune an entire (100%) block. Although it achieves profound throughput (Figure 2a), it is less flexible to scale up the pruning ratio (Figure 2b). In parallel, SliceGPT (Ashkboos et al., 2024) evenly prunes the matrix weights of each block. As the significance of each unit varies (Zhang et al., 2022), the ideal pruning ratio distributed for each unit might vary. Hence, such a prescribed proposal might be less flexible to scale up and limits the model efficiency considering performances.

To address the two aforementioned issues, we propose a hierarchical strategy to simultaneously target different structure units with a dynamic pruning ratio. In the first stage, we allocate the pruning proportion for each block by considering their importance. Specifically, we rank each block by measuring their interactions within the LLMs. Subsequently, in a transformer block, we dispatch the pruning load to different structure units considering their entropy. We use a simple clustering approach to mitigate the computation complexity. Our approach not only ensures significant improvements in the inference speed of the pruned model but also maximally preserves the original model's performance. Our contributions are summarized as follows:

- We propose a structured pruning framework to simultaneously target different structure units of LLMs, which balances the trade-off between inference speed and model performance.

- We introduce a hierarchical strategy to prune different LLM modules with a dynamic ratio. In the first stage, we determine the pruning proportion of each block by

measuring their interactions within the LLMs. We then dispatch the pruning load to the structured units within the block by entropy consideration.

- Our method has been rigorously validated across different benchmarks, demonstrating exceptional performance without relying on any fine-tuning or retraining.

## 2. Related Work

To reduce the inference cost of large language models and increase their practical applications, numerous recent studies have focused on model compression techniques. there have been many recent works on compressing models. A majority of model compression techniques fall into one of three categories: distillation (Gu et al., 2023; Agarwal et al., 2023; Ko et al., 2024; Padmanabhan et al., 2024), pruning and quantization (Frantar et al., 2022; Chee et al., 2024; Egiazarian et al., 2024; Zhang et al., 2024a). In the paper, we specifically focus on the pruning of the LLMs. Pruning can be categorized into unstructured pruning, semi-structured pruning and structured pruning.

**Unstructured pruning and semi-structured pruning.** Unstructured pruning involves removing individual elements from the weight matrix without regard to the network's structure, which allows the model to achieve a high degree of sparsity. However, this approach does not alter the matrix size and the inference speed remains nearly identical to that of the original model. To address this limitation, unstructured pruning can be combined with $N : M$ sparsity (Mishra et al., 2021), a technique known as "semi-structured pruning." In $N : M$ sparsity, at most N out of every M contiguous weights are allowed to remain non-zero. SparseGPT (Frantar & Alistarh, 2023) takes a different approach by employing the OBS technique (Hassibi et al., 1993) for pruning GPT-family models, marking it as the first method capable of effectively pruning LLMs with 10-100 billion or more parameters. RIA (Zhang et al., 2024b), another semi-structured pruning method, also considers both weights and activation. However, unlike Wanda, it proposes using the relative magnitude of the weights rather than their absolute values. Although semi-structured pruning can largely preserve model performance at high sparsity, its acceleration benefits are still heavily dependent on NVIDIA Ampere GPU architecture (Mishra et al., 2021).

**Structured pruning.** Unlike unstructured pruning, which removes individual elements without regard to the network's structural integrity, structured pruning specifically targets and removes entire components or modules within LLMs. LLM-Pruner (Ma et al., 2023) is a structured pruning technique that assesses the importance of module groups in LLMs by analyzing gradients and activations, systematically eliminating less significant components. SliceGPT (Ashkboos et al., 2024) employs a structured pruning strategy

similar to weight slicing (Li et al., 2021; 2022), by introducing embedding dimensionality reduction matrices for LLMs. These matrices facilitates the identification and elimination of less critical component. FLAP (An et al., 2024) introduces the concept of "fluctuation measurement" to assess the importance of rows&columns in the weight matrix. BlockPrune (Zhong et al., 2024) is a dynamic pruning method that masks attention and MLP layers sequentially, recording the PPL results. The layer with the lowest PPL is removed, and the process continues until the target pruning ratio is achieved. SLEB (Song et al., 2024) also proposes an iteration pruning method to assess the impact of each block based on the token prediction results of the LLMs. In each iteration, the block with the least impact on the prediction results is removed. ShortGPT (Men et al., 2024) proposes a metric called Block Influence (BI), which quantifies the changes in the hidden state of each block, to measure of block importance.

Yet, existing pruning methods typically focus on a single structure unit within LLMs, but such single-structure-oriented pruning strategies struggle to effectively balance model performance and efficiency after pruning. Moreover, existing pruning methods typically assign the same pruning ratio to each block without accounting for the varying importance of individual blocks within the LLMs. In contrast, we consider information entropy as a pruning criterion, enabling a unified pruning approach across all structure units with a dynamic ratio. Our approach not only maximizes the retention of the model's original performance but also significantly enhances inference speed.

## 3. Method

In this section, we propose a systematic pruning framework grounded in information-theoretic principles which simultaneously targets all the LLM modules. In Sec. 3.1, we first introduce the metric of transfer entropy to analyze the interaction among blocks in LLMs. This enables us to dynamically determine the pruning ratio for each block, and perform pruning on coarse-grained structure units such as blocks and layers in Sec. 3.2. Then, we further allocate the pruning load within each block and perform fine-grained pruning of the weight matrix rows&columns based on the information entropy of individual structure units in Sec. 3.3. Finally, in Sec. 3.4, our method use the bias compensation to enhance the performance of the pruned model without the need of post-training or any fine-tuning.

### 3.1. Quantifying the Importance of Blocks

Since the functionalities of network units would vary (Zhang et al., 2022), we propose to assign a pruning proportion to each block according to their role in the LLMs. We tend to assign a larger pruning proportion to the uninformative

blocks and preserve the essential blocks. That is, we want to formulate a metric to measure the importance of a transformer block.

**Measuring interactions of blocks.** We rank the blocks by capturing their complex interactions within the LLM. Intuitively, if the model output is highly correlated with a block, this block is identified as important; otherwise uninformative. To this end, we resort to the concept of transfer entropy (TE) (Schreiber, 2000), which quantifies the information transfer between two network components. Transfer entropy is formally defined as:

$$\text{TE} = H(\mathbf{X}_{out}) - H(\mathbf{X}_{out}|\text{Mask}\{block_i\}), \quad (1)$$

where $H(\mathbf{X}_{out})$ represents the original entropy of the final output layer, and $H(\mathbf{X}_{out}|\text{Mask}\{block_i\})$ denotes the entropy of final output layer after masking the $i^{th}$ block. Specifically, when a block is masked, Equation (1) provides a comprehensive evaluation of the impact on the hidden state of the LLM output. Thus, TE can measure the significance of $block_i$ on the model output and thus reflect its importance, i.e., a block is identified as important if the TE metric changes drastically. Consequently, we use the TE metric to guide the pruning proportion allocated to the blocks. Similarly, Equation (1) can be applied to attention or MLP layers. Figure 3a and Figure 3b illustrate the transfer entropy w.r.t. each block and layer of the LLaMA2-7B model. The transfer entropy of blocks varies dramatically across different orders of magnitude. Hence, rather than a static prescribed ratio, it is necessary to assign a pruning ratio to each block based on its dynamic interaction within the LLM.

**Entropy quantification.** For a distribution $p(x)$, its entropy can be written as:

$$H(\mathbf{X}) = -\int p(x) \log p(x), x \in \mathbf{X}, \quad (2)$$

where $x$ represents the hidden state in the feature space of a transformer unit $\mathbf{X}$. In prior work, such as (Sirignano & Spiliopoulos, 2020; Sun et al., 2022), it is assumed that $p(x)$ follows a Gaussian distribution, i.e. $\mathbf{X} \sim N(\mu, \sigma^2)$. Therefore, Equation (2) can be rewritten as:

$$\begin{aligned} H(\mathbf{X}) &= -E[\log N(\mu, \sigma^2)] \\ &= -E\left[\log\left[(2\pi\sigma^2)^{-1/2}\exp\frac{(x-\mu)^2}{2\sigma^2}\right]\right] \\ &= \log\sigma + \frac{1}{2}\log(2\pi) + \frac{1}{2}, \end{aligned} \quad (3)$$

where $\sigma$ is the standard deviation of $\mathbf{X}$.

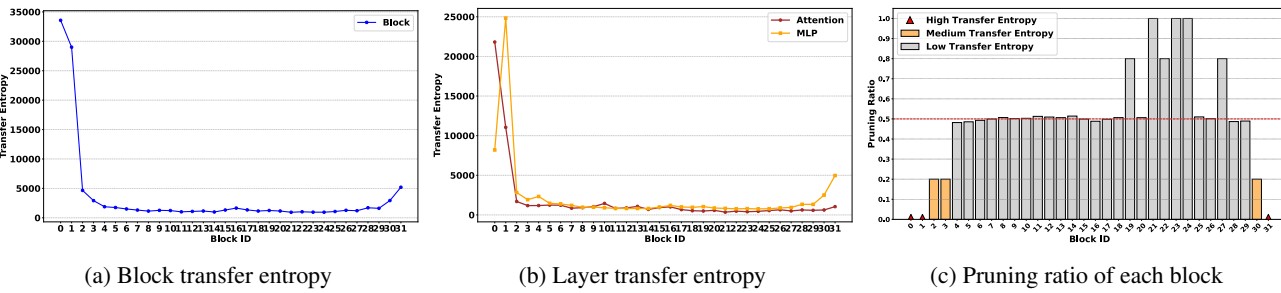

(a) Block transfer entropy      (b) Layer transfer entropy      (c) Pruning ratio of each block

*Figure 3.* The transfer entropy of LLaMA2-7b's block and layer. (a) We sequentially mask each block of the LLaMA2-7B model and compute the transfer entropy of the output layer for each masked configuration. (b) We sequentially mask each layer of the LLaMA2-7B model and calculate the transfer entropy for each corresponding layer. (c) The pruning ratio of each block using Algorithm 1.

### 3.2. Block-wise Pruning Ratio

**Grouping blocks based on TE.** We store the transfer entropy associated with each block (Figure 3a) in set $\mathcal{S}$ and divide them into three sub-sets: $\mathcal{S}_L$, $\mathcal{S}_M$ and $\mathcal{S}_H$ standing for low, medium and high TE set, based on the statistic of the set $\mathcal{S}$:

$$\mathcal{S} = \begin{cases} \mathcal{S}_L = \{s_i | s_i < T_l\}, \\ \mathcal{S}_M = \{s_i | T_l \leq s_i \leq T_h\}, \\ \mathcal{S}_H = \{s_i | s_i > T_h\}. \end{cases} \quad (4)$$

where $s_i$ represents the transfer entropy of $i^{th}$ block, $T_l = \mu_s - \alpha \cdot \sigma_s$ and $T_h = \mu_s + \alpha \cdot \sigma_s$ are two thresholds with the scaling factor $\alpha$. Here $\mu_s = \frac{1}{N}\sum_{i=1}^{N} s_i$ and $\sigma_s = \sqrt{\frac{1}{N}\sum_{i=1}^{N}(s_i - \mu)^2}$ are the mean and the standard deviation of the set $\mathcal{S}$, $N$ represents the number of blocks in a LLM. Similarly, we sort the transfer entropy of the attention and MLP layers (Figure 3b) in set $\mathcal{A}$.

**Hybrid pruning strategy.** Let $\mathcal{P}$ denote the set of pruning ratios. We allocate the pruning ratio to each block according to their importance in Equation (4): $\mathcal{S} \mapsto \mathcal{P}$. Since our pruning strategy involves multiple LLM modules simultaneously, we have five pruning types in general: a) $\mathcal{P}_Z$: blocks are preserved with zero pruning ratio; b) $\mathcal{P}_F$: blocks are fully pruned; c) $\mathcal{P}_E$: attention or MLP layers are completely pruned; d) $\mathcal{P}_D$: blocks are pruned with a lower pruning ratio; e) $\mathcal{P}_R$: the resting blocks with an adjusted ratio.

**Block-wise pruning.** We preserve the blocks in $\mathcal{S}_H$ as they are the essential blocks in LLM identified by transfer entropy, i.e., $\mathcal{S}_H \mapsto \mathcal{P}_Z$. The blocks in $\mathcal{S}_M$ would be assigned with a pruning ratio $p_{low}$ lower than the target ratio $p_{target}$, as they are the relatively essential blocks in LLM, i.e., $\mathcal{S}_M \mapsto \mathcal{P}_D$. To accommodate the pruning requirement, there would be a leftover $p_{leftover}$ that is redistributed to the blocks in $\mathcal{S}_L$:

$$p_{leftover} = |\mathcal{S}_H| \times p_{target} + |\mathcal{S}_M| \times (p_{target} - p_{low}), \quad (5)$$

where $|\cdot|$ represents the operation to obtain the number of elements in a set.

**Depth-First Search compensation process.** To compensate for the leftover, we should increase the pruning ratios of other blocks. As Figure 5 shown, initially, we select the top $|\mathcal{S}_H|$ blocks with the least TE from set $\mathcal{S}_L$ since they have least impact on the hidden state of LLM and place them into set $\mathcal{F}$, assigned with the pruning ratio 1 (i.e., completely pruned) and mapped to $\mathcal{P}_F$. The blocks assigned a pruning ratio of 1 have their corresponding layers completely removed. Therefore, these layers in set $\mathcal{A}$ will be classified as unavailable part. Next, the top $|\mathcal{S}_M|$ layers with the least TE are selected from the available part from set $\mathcal{A}$. These layers will be completely pruned, and their corresponding blocks are divided to the extensively pruned set $\mathcal{E}$, with their pruning ratio increased to $1 - p_{low}$ mapped to $\mathcal{P}_E$. For the remaining blocks that have not yet been assigned an initial pruning ratio, we incorporate them into set $\mathcal{R}$. Finally, we need to select according blocks from $\mathcal{F}$ and $\mathcal{E}$ to compensate for the leftover. How to select according blocks can be reformulated as the following optimization problem:

$$\arg \min_{\mathcal{L} \subseteq \mathcal{F}, \mathcal{M} \subseteq \mathcal{E}} \|p_{compensation} - p_{leftover}\|$$
$$s.t. : p_{compensation} \geq p_{leftover}, \quad (6)$$

where

$$p_{compensation} = |\mathcal{L}|(1 - p_{target}) + |\mathcal{M}|(1 - p_{target} - p_{low}). \quad (7)$$

$\mathcal{L}$ and $\mathcal{M}$ represent subsets of blocks selected from sets $\mathcal{F}$ and $\mathcal{E}$, respectively. This problem can be efficiently solvable using a Depth-First search algorithm. This optimization problem may yield multiple solutions $\{\mathcal{L}^*, \mathcal{M}^*\}$. To achieve optimal inference speed, we select the solution with the maximum $|\mathcal{L}^*|$ as the final result. The unselected blocks in $\mathcal{F}$ and $\mathcal{E}$ are returned to $\mathcal{R}$, with their pruning ratios reassigned accordingly.

**Dynamic ratio assigning**. For the remaining blocks in

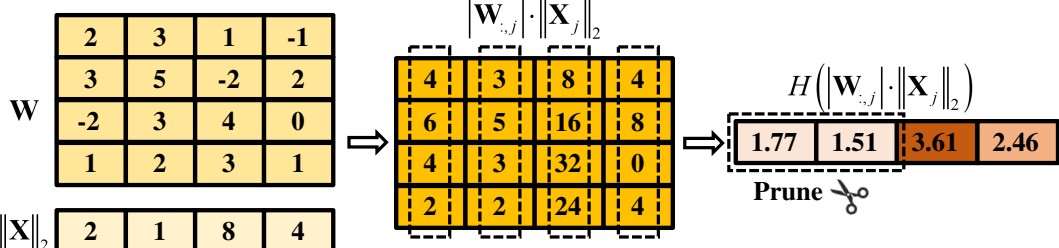

*Figure 4.* Illustration of our proposed method based on information entropy. (Left): A weight matrix $\mathbf{W}$ and the $L_2$ norm of the activation value $\|\mathbf{X}\|_2$. (Middle): Multiply the absolute value of weight matrix by the $L_2$ norm of the activation value. (Right): Calculate the information entropy of each column, perform clustering using the $K$-means algorithm, and represent different clusters with distinct colors, where lighter colors indicate lower information entropy for the corresponding cluster.

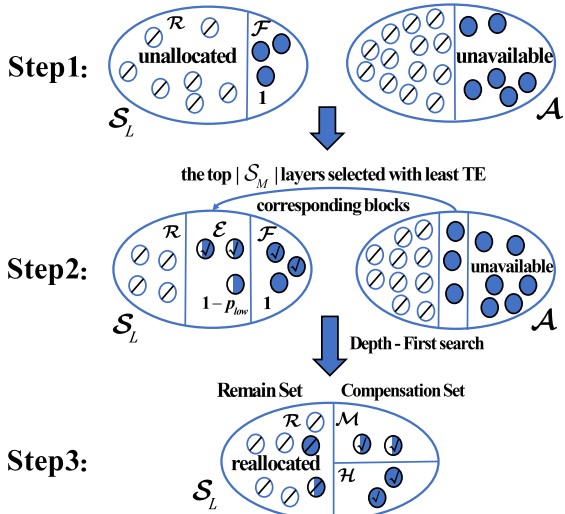

**Step1:**

**Step2:**

the top $|\mathcal{S}_M|$ layers selected with least TE corresponding blocks

Depth – First search

Remain Set     Compensation Set

**Step3:**

*Figure 5.* Compensation process. Each ○ represents a block, and its fill level visually corresponds to the pruning ratio. ⊘ represents a block that has not yet been assigned a pruning ratio.

$\mathcal{R}$, we dynamically allocate these ratios based on transfer entropy. The pruning ratio for each block $g$ is calculated as follows:

$$p_g = \frac{\omega_g}{\frac{1}{G}\sum_{g=1}^{G}\omega_g} \times \hat{p}_{target}, \qquad (8)$$

where

$$\hat{p}_{target} = (G \cdot p_{target} - p^*_{compensation} + p_{leftover})/G,$$

$p^*_{compensation}$ represents the result obtained by substituting the optimized solution $\{\mathcal{L}^*, \mathcal{M}^*\}$ into Equation (6). $G$ is the number of blocks in $\mathcal{R}$, weight $\omega_g = \left(\frac{\bar{s}}{s_g}\right)^{\beta}$, $\beta$ is a scaling factor and average information entropy $\bar{s} = \left(\frac{1}{G}\sum_{g=1}^{G}s_g\right)$. We summarize this assigning block-wise pruning ratio process, as shown in Algorithm 1.

---

**Algorithm 1** Block-wise pruning ratio assignment

**Input:** an original LLM, target pruning ratio $p_{target}$.
**Output:** the pruning ratio for each block.
Calculate the transfer entropy of block and layer:
**for** $i = 0$ **to** $N - 1$ **do**
    Store the TE values corresponding to layers and blocks in the set $\mathcal{A}$ and set $\mathcal{S}$ using Eq. (1).
**end for**
Divide $\mathcal{S}$ into three subsets $\mathcal{S}_L$, $\mathcal{S}_M$ and $\mathcal{S}_H$ using Eq. (4).
$\mathcal{S}_H \leftarrow 0\%$, $\mathcal{S}_M \leftarrow p_{low}$, calculate $p_{leftover}$ using Eq. (5).
$\mathcal{F} \leftarrow \operatorname{argmin}_{|\mathcal{S}_H|}\{\mathcal{S}_L\}$, $\mathcal{E} \leftarrow \operatorname{argmin}_{|\mathcal{S}_M|}\{A\}$
$\mathcal{F} \leftarrow 1$, $\mathcal{E} \leftarrow 1 - p_{low}$.
Depth-First Search to solve this optimization problem 6.
Assign pruning ratios to the remaining blocks using Eq. (8).

---

### 3.3. Row&Column-wise Pruning

**Distributing the pruning load.** After determining the appropriate pruning ratios for each block, we sign the pruning load to the structured units within the block, i.e. finer-grained structure pruning by eliminating entire rows&columns in weight matrices. Building on the approach proposed in (Sun et al., 2023), we propose an importance metric driven by information entropy that jointly weights $\mathbf{W}$ and the activation value $\mathbf{X}$. Taking the $j$-th structured unit(column) in $\mathbf{W}$ as an example, the importance score $\mathbf{IS}_j$ is calculated as follows:

$$\mathbf{IS}_j = H(|\mathbf{W}_{:,j}| \cdot \|\mathbf{X}_j\|_2), \qquad (9)$$

where $H(\cdot)$ denotes the calculation of the information entropy, $|\cdot|$ represents the absolute value operator, and $\|\mathbf{X}_j\|_2$ is the $l_2$-norm of $j$-th channel across $N \times L$ different tokens. Upon obtaining the information entropy for each column in the weight matrix, we apply the $K$-means algorithm (MacQueen et al., 1967) to cluster the units based on their entropy values and prune the cluster with the lowest average information entropy. Figure 4 illustrates the process of assessing

*Table 1.* The perplexity results of pruned LLaMA2, LLaMA3 and Vicuna models.

| METHOD | PRUNING UNIT | PRUNING RATIO | LLaMA2-7B | LLaMA2-13B | LLaMA2-70B | LLaMA3-8B | LLaMA3-70B | VICUNA-7B | VICUNA-13B |
|---|---|---|---|---|---|---|---|---|---|
| DENSE | - | 0% | 5.47 | 4.88 | 3.31 | 5.53 | 2.85 | 6.78 | 5.94 |
| SPARSEGPT | 2:4 | 50% | 9.33 | 7.90 | 5.74 | 19.99 | 6.73 | 10.31 | 9.22 |
| WANDA | 2:4 | | 11.00 | 8.23 | 5.26 | 10.89 | 7.77 | 12.41 | 8.96 |
| FLAP | ROW&COLUMN | | 9.13 | 7.31 | 4.89 | 10.64 | 5.99 | 11.21 | 10.13 |
| SLICEGPT | ROW&COLUMN | | 8.63 | 7.43 | 5.42 | 13.08 | 9.01 | 9.93 | 11.32 |
| LLM-PRUNER | ROW&COLUMN | 30% | 17.86 | 19.38 | - | 22.88 | - | 21.17 | 14.27 |
| BLOCKPRUNER | LAYER | | 15.06 | 8.52 | 6.27 | 26.91 | 10.09 | 13.90 | 9.93 |
| SLEB | BLOCK | | 27.45 | 23.48 | 5.93 | 29.17 | 9.08 | - | - |
| OURS | ALL | | 8.71 | **7.03** | **4.77** | **10.53** | **5.83** | 9.90 | **8.54** |
| FLAP | ROW&COLUMN | | 14.49 | 9.60 | 5.71 | **14.13** | 7.14 | 14.42 | 11.65 |
| SLICEGPT | ROW&COLUMN | | 12.79 | 10.61 | 7.08 | 20.1 | 13.46 | 14.09 | 19.79 |
| LLM-PRUNER | ROW&COLUMN | 40% | 46.33 | 43.10 | - | 56.28 | - | 43.52 | 27.24 |
| BLOCKPRUNER | LAYER | | 32.39 | 13.91 | 8.42 | 74.75 | 16.34 | 29.94 | 16.64 |
| SLEB | BLOCK | | 45.36 | 27.49 | 7.57 | 78.66 | 13.39 | - | - |
| OURS | ALL | | **12.55** | **8.85** | **5.64** | 14.72 | **6.93** | **13.91** | **10.54** |
| FLAP | ROW&COLUMN | | 25.98 | 15.25 | 7.13 | 23.12 | **8.76** | 29.23 | 18.91 |
| SLICEGPT | ROW&COLUMN | | 21.08 | 17.51 | 10.75 | 35.64 | 28.69 | 23.01 | 45.01 |
| LLM-PRUNER | ROW&COLUMN | 50% | 286.07 | 89.31 | - | 185.86 | - | 189.08 | 70.37 |
| BLOCKPRUNER | LAYER | | 58.86 | 29.13 | 12.81 | 479.23 | 61.47 | 80.16 | 36.07 |
| SLEB | BLOCK | | 106.02 | 51.56 | 11.58 | 390.56 | 24.13 | - | - |
| OURS | ALL | | **20.07** | **13.44** | **6.76** | **21.97** | 8.92 | **22.05** | **15.20** |

the importance of columns in a weight matrix using information entropy and subsequently removing the less important columns.

**Practical pruning process.** In practical pruning process, due to the attention head mechanism employed in LLMs, where each attention head corresponds to a subset of rows or columns, attention head are typically treated as the primary unit for pruning the attention layers. In the MLP layer, rows or columns are treated as the fundamental pruning units. The pruning procedure for different LLMs is provided in the Appendix A.

### 3.4. Bias Compensation

Bias compensation is a common compensatory mechanism used to recover the performance of models after pruning or quantization (Gong et al., 2024; Finkelstein et al., 2019; An et al., 2024). The compensation bias term for the $j$-th channel is computed as a weighted average across all samples, formulated as follows:

$$\hat{\bar{\mathbf{X}}}_j = \frac{\sum_{n=1}^{N} \sum_{l=1}^{L} \lambda_1(n)\lambda_2(l)\mathbf{X}_{n,l,j}}{\sum_{n=1}^{N} \sum_{l=1}^{L} \lambda_1(n)\lambda_2(l)}, \quad (10)$$

where the normalized weighting factors $\lambda_1(n)$ and $\lambda_2(l)$ modulate contributions along the sequence and channel dimensions respectively:

$$\lambda_1(n) = \frac{\sigma_1(n)}{\sum_{n=1}^{N} \sigma_1(n)},$$

$$\lambda_2(l) = \frac{\sigma_2(l)}{\sum_{l=1}^{L} \sigma_2(l)}. \quad (11)$$

Here, $\sigma_1$ denotes the standard deviation computed over the joint sequence-channel dimensions of $\mathbf{X}$, while $\sigma_2$ represents the standard deviation across the batch-channel dimensions. This weighted approach adaptively prioritizes

dimensions with higher activation variability through $\sigma_1$ and $\sigma_2$, thereby more effectively preserving critical features and maintaining the original activation statistics.

Once the rows&columns in the weight matrix that require pruning are identified, we can employ a binary mask matrix $\mathbf{M}$ to indicate the locations of pruning (0 for pruned, 1 for retained). According to the binary mask matrix $\mathbf{M}$ and weighted average $\hat{\bar{\mathbf{X}}}$, we can calculate the corresponding bias term $\mathbf{b}_0$ as follows:

$$\mathbf{b}_0 = \mathbf{W}((\mathbf{1} - \mathbf{M}) \odot \hat{\bar{\mathbf{X}}}). \quad (12)$$

where $\mathbf{1}$ denotes an all-ones matrix and $\odot$ represents Hadamard product. Subsequently, we employ bias to compensate for the impact of pruning on the model's output, as shown in the following equation:

$$\mathbf{W}\mathbf{X} \approx (\mathbf{M} \odot \mathbf{W})\mathbf{X} + \mathbf{b}_0 \quad (13)$$

## 4. Experiment

### 4.1. Experimental Setup

**Setup.** In our experiment, we apply our method to various large language models (LLMs), including LLaMA2 (Touvron et al., 2023), LLaMA3 (Meta, 2024), and Vicuna-7B/13B (Chiang et al., 2023), all of which are available through the Hugging Face Transformers library (Wolf, 2019). All experiments are conducted on NVIDIA A800 GPUs with 80GB memory.

**Datasets and Evaluation.** For calibration data, we randomly select 128 samples from the WikiText2 training dataset (Merity et al., 2016). Since pruning can affect the model's ability, it is crucial to measure its impact using a reliable evaluation metric. Perplexity serves as the primary metric for evaluating model performance, with lower perplexity indicating better performance. Therefore, we focus

*Table 2.* Zero-shot performance of the compressed LLaMA2-7B. Bold results highlight the best performance.

| LLaMA2-7B | PRUNING UNIT | SPARSITY | WINOGRANDE | PIQA | HELLASWAG | ARC-E | ARC-C | AVG |
|---|---|---|---|---|---|---|---|---|
| DENSE | - | 0% | 68.98 | 79.11 | 76.00 | 74.58 | 46.25 | 68.98 |
| SPARSEGPT | 2:4 | 50% | 61.03 | 68.77 | 52.80 | 56.69 | 30.63 | 53.99 |
| WANDA | 2:4 | | 60.93 | 69.80 | 54.16 | 58.08 | 30.89 | 54.77 |
| FLAP | ROW&COLUMN | | 61.25 | 70.67 | 56.79 | 54.84 | 32.08 | 55.12 |
| SLICEGPT | ROW&COLUMN | | **62.67** | 64.80 | 49.19 | 50.38 | 31.48 | 51.70 |
| LLM-PRUNER | ROW&COLUMN | 30% | 54.93 | **72.85** | 57.79 | 53.16 | 32.51 | 54.24 |
| BLOCKPRUNER | LAYER | | 57.62 | 70.51 | 56.62 | 50.51 | 30.97 | 53.24 |
| SLEB | BLOCK | | 52.33 | 69.80 | 51.49 | 51.85 | 30.03 | 51.10 |
| OURS | ALL | | 60.77 | 70.95 | **59.18** | **58.16** | **34.22** | **56.65** |
| FLAP | ROW&COLUMN | | 56.51 | **67.57** | 49.00 | 37.71 | 27.30 | 47.62 |
| SLICEGPT | ROW&COLUMN | | **57.22** | 58.22 | 39.49 | 43.39 | 27.05 | 45.07 |
| LLM-PRUNER | ROW&COLUMN | 40% | 52.80 | 66.32 | 43.44 | 39.23 | 29.95 | 46.34 |
| BLOCKPRUNER | LAYER | | 53.99 | 63.00 | 45.70 | 42.51 | 29.01 | 46.84 |
| SLEB | BLOCK | | 51.38 | 64.85 | 42.75 | 44.95 | 25.85 | 45.95 |
| OURS | ALL | | 56.67 | 66.10 | **49.00** | **49.83** | **30.38** | **50.39** |
| FLAP | ROW&COLUMN | | 53.35 | **62.40** | 40.31 | 36.53 | 25.09 | 43.53 |
| SLICEGPT | ROW&COLUMN | | 53.43 | 53.81 | 32.64 | 34.89 | 23.63 | 39.68 |
| LLM-PRUNER | ROW&COLUMN | 50% | 50.59 | 56.96 | 31.23 | 31.23 | 26.62 | 39.32 |
| BLOCKPRUNER | LAYER | | 53.35 | 58.16 | 36.32 | 37.50 | 26.11 | 42.28 |
| SLEB | BLOCK | | 50.75 | 57.34 | 34.41 | 33.92 | 26.28 | 40.54 |
| OURS | ALL | | **53.83** | 61.81 | **40.46** | **43.94** | **27.56** | **45.52** |

on perplexity and utilize the WikiText2 test dataset to evaluate the perplexity of the model after pruning with different pruning methods. Furthermore, to further validate the efficacy of our method, we report the Zero-Shot accuracy on five benchmark datasets: PIQA (Bisk et al., 2020); Wino-Grande (Sakaguchi et al., 2021); HellaSwag (Zellers et al., 2019); ARC-e and ARC-c (Clark et al., 2018).

**Baselines.** We compare our proposed approach with pruning methods designed for individual structure units: LLM-Prune (Ma et al., 2023), FLAP (An et al., 2024) (utilizing bias compensation) and SliceGPT (Ashkboos et al., 2024) prune the rows&columns of the weight matrices. Block-Prune (Zhong et al., 2024) and SLEB (Song et al., 2024) separately focus on pruning the layers (attention&MLP) and the blocks in LLMs.

### 4.2. Pruning Results on LLMs

**Generation Tasks.** We begin by presenting the WikiText2 performance results for the LLaMA2, LLaMA3, and Vicuna models at three distinct pruning ratios (30%, 40%, and 50%) in Table 1. As illustrated in Table 1, our approach, which prunes across all structure units, consistently achieves superior perplexity performance compared to methods that prune based on rows and columns, such as SliceGPT, FLAP, and LLM-Pruner, across most LLMs. Notably, at a pruning ratio of 30%, our method outperforms semi-structured pruning techniques, including SparseGPT and Wanda, across LLMs.

**Zero-shot Tasks.** Table 2 presents the zero-shot accuracies of the pruned LLaMA2-7B model across five different tasks, evaluated using various pruning methods. As expected, at the same pruning ratio, our method consistently

outperforms SliceGPT, FLAP, and LLM-Pruner, even after removing a significant number of blocks and layers. Compared to pruning methods that focus primarily on removing blocks or layers, such as SLEB and BlockPruner, our method achieves significantly better results across all zero-shot tasks. Even compared to semi-structured pruning methods such as SparseGPT and Wanda, our approach delivers superior performance at the 30% pruning ratio. These results demonstrate that our method not only effectively reduces the model size but also maximally preserves the original performance of the LLMs. For further details, additional accuracy results for other LLMs across these tasks can be found in Appendix B.

### 4.3. Inference Speed

In Table 3, we provide a detailed comparison of the inference speed for generation the sequences of length 128 (batch size of 1) under different pruning methods. All experiments are conducted on NVIDIA A800 GPUs and we observe that unstructured pruning methods like Wanda and SparseGPT do not significantly improve inference speed. Compared to row&column pruning methods such as SliceGPT and FLAP, our approach achieves a significant improvement in inference speed while requiring less memory overhead. Furthermore, as the pruning ratio increases, the improvement in inference speed achieved by our method becomes more pronounced. This is because, at higher pruning ratios, our approach tends to remove more blocks and layers within LLMs. For example, when pruning LLaMA2-7B model at a 50% ratio, we remove three blocks and three layers, respectively, further enhancing inference speed.

*Table 3.* Inference speed of LLaMA2-7B/13B compared with different pruning methods.

| Method | Sparsity | Memory | Tokens/s |
|---|---|---|---|
| LLaMA2-7B | 0% | 12989MB | 29.86 |
| Wanda | 50% | 12989MB | 28.60 |
| SparseGPT | | 12989MB | 30.34 |
| SliceGPT | | 10118MB | 33.39 |
| FLAP | 30% | 9296MB | 32.81 |
| Ours | | 9297MB | 34.88 |
| SliceGPT | | 7172MB | 34.38 |
| FLAP | 50% | 6833MB | 34.39 |
| Ours | | 6800MB | 38.78 |
| LLaMA2-13B | 0% | 25114MB | 25.64 |
| Wanda | 50% | 25114MB | 26.85 |
| SparseGPT | | 25114MB | 25.35 |
| SliceGPT | | 19631MB | 29.11 |
| FLAP | 30% | 17767MB | 26.90 |
| Ours | | 17701MB | 29.22 |
| SliceGPT | | 12995MB | 29.79 |
| FLAP | 50% | 12922MB | 27.91 |
| Ours | | 12632MB | 31.80 |

## 4.4. Baseline Bias Compensation

Our approach leverages bias compensation, which eliminates the need for retraining or fine-tuning. To evaluate its effectiveness, we conduct experiments on the LLaMA2-13B model with different pruning ratio. The results shown in Figure 6, highlight that bias compensation effectively restores the model's performance after pruning. As the pruning ratio increases, the impact of bias compensation becomes more pronounced, emphasizing its importance in maintaining performance. Notably, even without the use of bias compensation, our method outperforms other approaches without bias compensation in terms of PPL.

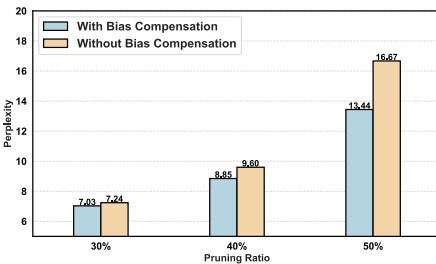

*Figure 6.* Performance comparison of the LLM with and without bias compensation at various pruning ratios.

## 4.5. Robustness to Calibration Samples

The effectiveness of different pruning methods can be affected by the number of samples in the calibration dataset. Therefore, this makes it critical to verify whether our method remains robust and outperforms other pruning methods with calibration datasets of varying samples. In the section, we select SliceGPT as the comparison method because, as shown in Table 1, it exhibits the second-best performance when pruning LLaMA2-7B at a 50% ratio. As Figure 7 shown,

our method consistently outperform SliceGPT across the different number of samples. Moreover, our method exhibits the smallest perplexity variation as the number of samples increasing, making it more robust compared to SliceGPT.

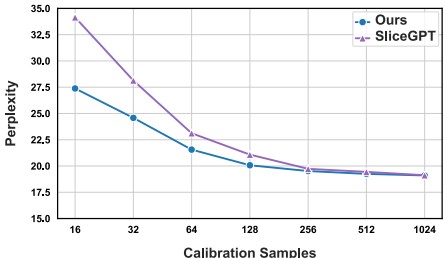

*Figure 7.* Robustness to calibration samples in the LLaMA2-7B with 50% pruning ratio.

## 4.6. Dependency on Calibration Dataset

Different calibration datasets can affect the performance of the pruned model. Therefore, it is important to evaluate how different calibration datasets affect the performance of different pruning methods. We use C4 and WikiText2 training datasets as different calibration datasets. Then, we utilize different structured pruning methods to evaluate the perplexity of the pruned LLaMA2-7B with 50% pruning ratio on both C4 and WikiText2 test datasets. In Figure 8, regardless of the choice of calibration dataset, the performance variation of our method is much smaller than that of other methods. This indicates that our approach does not rely on the selection of the calibration dataset compared to other methods. Furthermore, regardless of the chosen test dataset, our method consistently outperforms other pruning methods.

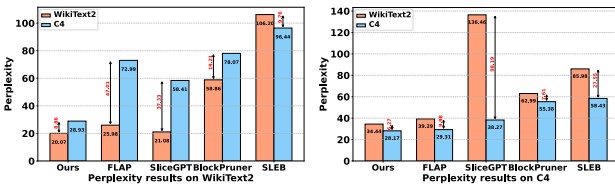

(a) Perplexity results on Wiki-Text2.    (b) Perplexity results on C4.

*Figure 8.* Perplexity results of LLaMA2-7B with 50% pruning ratio on the different calibration dataset.

## 4.7. Ablation Studies

In this section, we conduct comprehensive ablation studies to evaluate the effectiveness of our proposed method. Specifically, we compare various evaluation criteria for quantifying module interactions and different search methods, analyze the performance differences between multi-structured units and single-structured unit pruning, and assess the impact of dynamic versus static pruning ratio allocation strategies on the performance of the pruned model. The detailed results of the ablation studies are provided in Appendix C.

# 5. Conclusion

In this paper, we propose a a structured pruning framework to simultaneously target different structure units of LLMs with dynamic ratios. Our method can effectively balance the trade-off between model performance and inference speed. We validate our proposed method across multiple LLMs and performe a comprehensive comparison with various structured pruning methods. The experiment results demonstrate that our method can effectively maintain model performance after pruned and enhance inference efficiency.

# Acknowledgments

This work is partially supported by CAS Pioneer Hundred Talents Program. The authors thank Pumeng Lyu for his helpful discussion.

# Impact Statement

This paper presents work whose goal is to advance the field of Machine Learning. There are many potential societal consequences of our work, none which we feel must be specifically highlighted here.

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

# A. Implementation Details

## A.1. Detailed Pruning Process for LLMs Using the Multi-Head Attention Mechanism

The LLaMA2-7B, LLaMA2-13B, and Vicuna models employ a multi-head attention mechanism (MHA), where each attention head is comprised of a set of rows or columns from the weight matrix, functioning as an independent computational unit. Directly pruning the rows or columns of the weight matrix in the attention layer can compromise the integrity of the attention heads. Thus, to address this, we treat entire attention heads as the fundamental pruning units to preserve the functionality and coherence of the attention mechanism.

We first prune the attention heads in the attention layer. As illustrated in Figure 9a, after multiplying the absolute value of the weight matrix by the $L_2$ norm of the activation value, we perform an averaging operation on the columns of each attention head within the output projection weight matrix $\mathbf{W}_o$, and then compute the information entropy for these averaged values. Based on the calculated information entropy, we cluster and sort the attention heads. The cluster with the smallest average information entropy are pruned. When the columns of $\mathbf{W}_o$ are pruned, the corresponding rows in $\mathbf{W}_q$ (query projection weight matrix), $\mathbf{W}_k$ (key projection weight matrix), and $\mathbf{W}_v$ (value projection weight matrix) should also be pruned accordingly, as illustrated in Figure 9b. To achieve the target pruning ratio for each block, the remaining portion is pruned in the MLP layer.

For the weight matrices in the MLP layer, we directly calculate the information entropy of the columns in $\mathbf{W}_{down}$ (down-projection weight matrix). After clustering and sorting the columns based on their information entropy, we prune the smallest average entropy within each cluster. Consequently, the corresponding rows in $\mathbf{W}up$ (up-projection matrix) and $\mathbf{W}gate$ (gate weight matrix) are also pruned to maintain structure consistency.

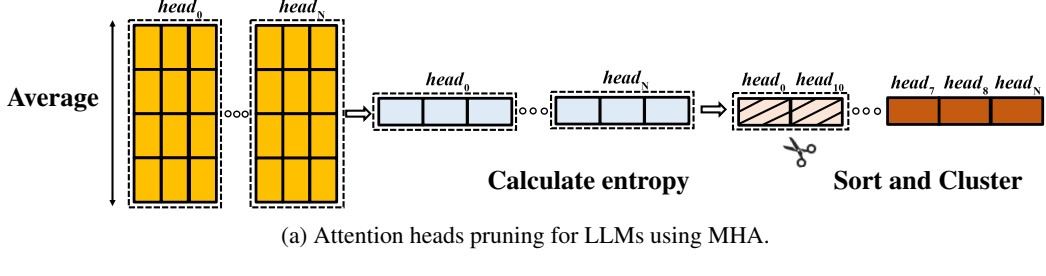

(a) Attention heads pruning for LLMs using MHA.

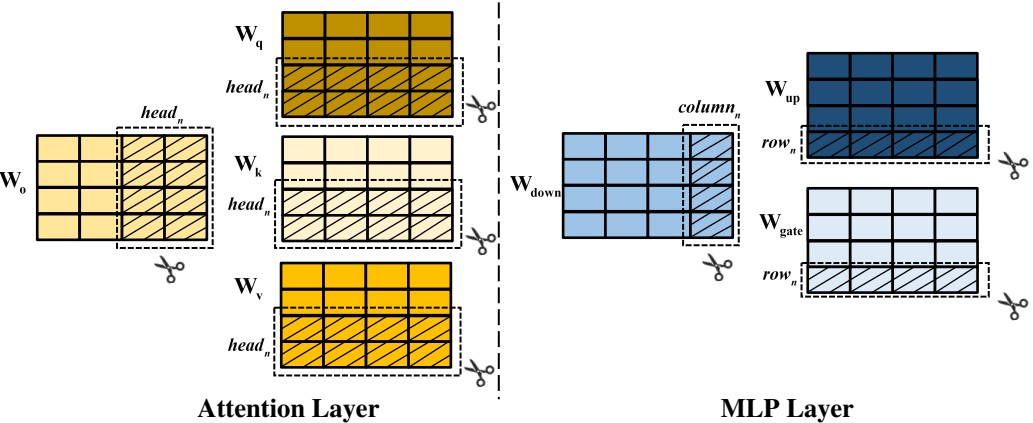

(b) Pruning process for LLMs using MHA.

*Figure 9.* The detailed pruning process for LLMs using MHA. (a) Attention head pruning. (b) The pruning process for the attention layer and MLP layer.

## A.2. Detailed Pruning Process for LLMs Using the Group-Query Attention Mechanism

Group-Query Attention mechanism (GQA) is a variant of the multi-head self-attention mechanism (MHA), designed to enhance computational efficiency by reducing computation and storage requirements. In GQA, the query heads are divided into $G$ groups, with each group sharing a single key and value head. LLaMA2-70B, LLaMA3-8B and LLaMA3-70B models use the GQA, therefore, different from Appendix A.1, we treat the group as the pruning unit as Figure 10a shown. Consistent with Figure 9a, we calculate the information entropy of each group within the $\mathbf{W}_o$, perform clustering, and prune the cluster with the lowest information entropy. When the groups of $\mathbf{W}_o$ are pruned, the corresponding groups in $\mathbf{W}_q$, and the corresponding heads in $\mathbf{W}_k$ and $\mathbf{W}_v$ should also be pruned accordingly, as shown in Figure 10b. In order to reach the desired pruning ratio for each block, the remaining portion is pruned in the MLP layer accordingly. For the weight matrix in the MLP layer, the pruning process is consistent with that described in Appendix A.1.

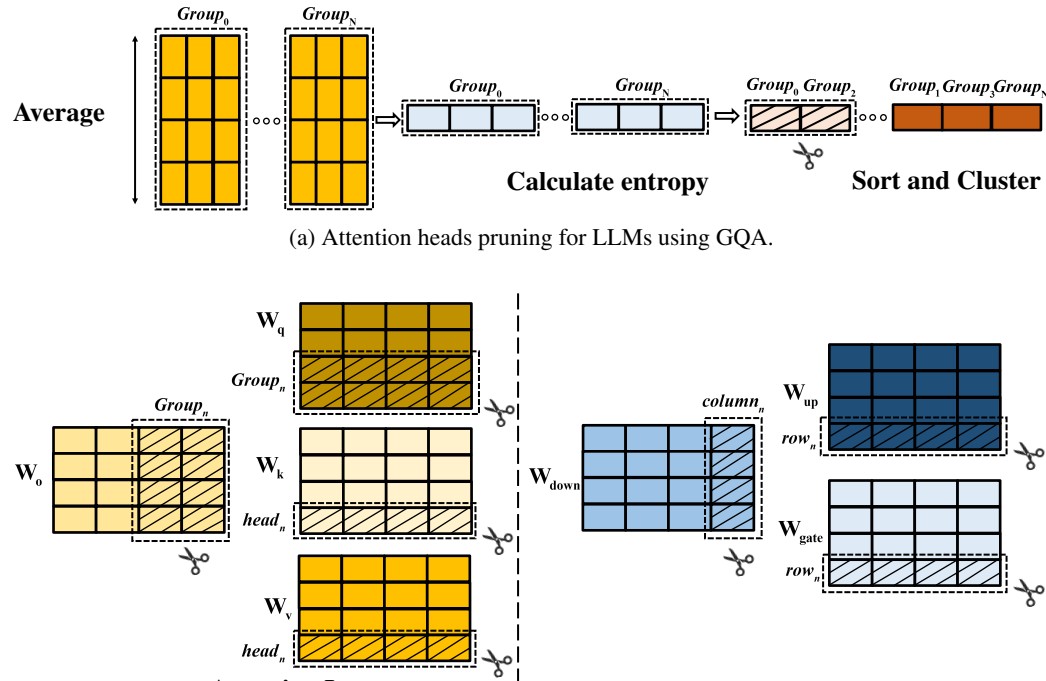

(a) Attention heads pruning for LLMs using GQA.

(b) Pruning process for LLMs using GQA.

*Figure 10.* The detailed pruning process for LLMs using GQA. Upper:Attention head pruning. Down: The pruning process for the attention layer and MLP layer.

# B. Additional Experiment Results

In this section, we provide the more detailed zero-shot tasks results of the different LLMs .

*Table 4.* Zero-shot performance of the compressed LLaMA2-13B. Bold results highlight the best performance.

| LLaMA2-13B | Pruning Unit | Sparsity | Winogrande | PIQA | Hellaswag | ARC-E | ARC-C | Avg |
|---|---|---|---|---|---|---|---|---|
| Dense | - | 0% | 72.30 | 80.52 | 79.37 | 77.53 | 49.06 | 71.75 |
| SparseGPT | 2:4 | 50% | 65.75 | 71.22 | 58.97 | 61.03 | 34.90 | 58.37 |
| Wanda | 2:4 | | 65.35 | 74.86 | 62.07 | 65.19 | 36.86 | 60.86 |
| SliceGPT | Row&Column | | 65.59 | 64.96 | 52.35 | 53.54 | 36.86 | 54.65 |
| LLM-Pruner | Row&Column | | 61.88 | 73.72 | 64.37 | 63.72 | 38.05 | 60.34 |
| BlockPrune | Layer | 30% | 63.22 | **75.08** | **65.28** | 66.37 | 40.02 | 61.99 |
| SLEB | Block | | 56.43 | 74.65 | 62.11 | 63.09 | 37.37 | 58.72 |
| Ours | All | | **66.69** | 73.61 | 64.83 | **67.68** | **41.47** | **62.85** |
| SliceGPT | Row&Column | | 60.69 | 59.30 | 42.50 | 56.44 | 34.13 | 47.30 |
| LLM-Pruner | Row&Column | | 54.70 | 68.23 | 50.50 | 49.45 | 31.40 | 50.85 |
| BlockPrune | Layer | 40% | 58.80 | **71.00** | **57.15** | 56.14 | **35.58** | 55.73 |
| SLEB | Block | | 55.49 | 70.24 | 54.08 | 54.25 | 32.34 | 53.27 |
| Ours | All | | **62.90** | 69.48 | 54.25 | **59.05** | 34.56 | **56.04** |
| SliceGPT | Row&Column | | 55.17 | 55.22 | 34.33 | 37.08 | 24.49 | 41.25 |
| LLM-Pruner | Row&Column | | 54.06 | 62.68 | 40.85 | 39.02 | 27.22 | 44.76 |
| BlockPrune | Layer | 50% | 54.62 | 61.48 | 45.08 | 41.71 | 28.16 | 46.02 |
| SLEB | Block | | 55.01 | 62.95 | 43.29 | 42.76 | 26.11 | 46.02 |
| Ours | All | | **55.72** | **65.72** | **47.30** | **45.54** | **29.10** | **48.68** |

*Table 5.* Zero-shot performance of the compressed LLaMA2-70B. Bold results highlight the best performance.

| LLaMA2-70B | Pruning Unit | Sparsity | Winogrande | PIQA | Hellaswag | ARC-E | ARC-C | Avg |
|---|---|---|---|---|---|---|---|---|
| Dense | - | 0% | 77.98 | 82.70 | 83.82 | 80.98 | 57.42 | 76.57 |
| SparseGPT | 2:4 | 50% | 74.82 | 78.62 | 76.11 | 77.86 | 52.47 | 71.97 |
| Wanda | 2:4 | | 74.66 | 79.49 | 75.86 | 76.73 | 51.45 | 71.64 |
| SliceGPT | Row&Column | | 73.09 | 71.55 | 62.72 | 73.95 | 49.23 | 66.10 |
| BlockPrune | Layer | | 70.40 | 78.02 | 72.50 | 69.78 | 43.09 | 66.75 |
| SLEB | Block | 30% | 69.53 | 77.80 | 73.57 | 72.01 | 44.03 | 67.38 |
| Ours | All | | **73.64** | **79.43** | **80.09** | **76.47** | **51.11** | **72.15** |
| SliceGPT | Row&Column | | 70.48 | 65.07 | 50.6 | 61.15 | 38.75 | 57.20 |
| BlockPrune | Layer | | 65.27 | 74.48 | 65.99 | 61.91 | 38.05 | 61.15 |
| SLEB | Block | 40% | 66.61 | 75.03 | 66.76 | 66.62 | 39.68 | 62.94 |
| Ours | All | | **72.30** | **76.71** | **77.26** | **71.09** | **47.95** | **69.06** |
| SliceGPT | Row&Column | | 63.61 | 58.11 | 39.45 | 45.75 | 30.03 | 47.39 |
| BlockPrune | Layer | | 60.38 | 69.15 | 58.49 | 51.56 | 34.04 | 54.72 |
| SLEB | Block | 50% | 61.25 | 70.29 | 57.67 | 58.00 | 35.58 | 56.55 |
| Ours | All | | **71.27** | **74.43** | **72.00** | **62.79** | **41.72** | **64.44** |

*Table 6.* Zero-shot performance of the compressed Vicuna-7B. Bold results highlight the best performance.

| VICUNA-7B | PRUNING UNIT | SPARSITY | WINOGRANDE | PIQA | HELLASWAG | ARC-E | ARC-C | AVG |
|---|---|---|---|---|---|---|---|---|
| DENSE | - | 0% | 68.38 | 78.02 | 73.77 | 71.30 | 45.82 | 67.65 |
| SPARSEGPT | 2:4 | 50% | 61.17 | 68.72 | 55.70 | 59.60 | 33.70 | 55.77 |
| WANDA | 2:4 | | 62.75 | 71.60 | 57.14 | 61.07 | 35.07 | 57.52 |
| SLICEGPT | ROW&COLUMN | | 59.83 | 63.87 | 48.99 | 49.24 | 31.74 | 50.73 |
| LLM-PRUNER | ROW&COLUMN | | 56.51 | 73.50 | 58.37 | 56.65 | 35.07 | 56.02 |
| BLOCKPRUNER | LAYER | 30% | 57.30 | **71.49** | 56.98 | 58.42 | 33.36 | 55.51 |
| OURS | ALL | | **64.48** | 70.18 | **60.43** | **59.89** | **37.29** | **58.45** |
| SLICEGPT | ROW&COLUMN | | 57.22 | 58.71 | 40.23 | 43.31 | 27.56 | 45.41 |
| LLM-PRUNER | ROW&COLUMN | | 51.62 | 67.85 | 47.60 | 46.97 | **32.59** | 49.32 |
| BLOCKPRUNER | LAYER | 40% | 53.43 | 63.55 | 45.97 | 49.37 | 28.84 | 48.23 |
| OURS | ALL | | **58.64** | **65.78** | **50.63** | **53.32** | 30.97 | **51.87** |
| SLICEGPT | ROW&COLUMN | | 50.28 | 53.26 | 32.67 | 34.76 | 23.98 | 38.99 |
| LLM-PRUNER | ROW&COLUMN | | 54.06 | 59.14 | 34.74 | 34.81 | 28.07 | 42.65 |
| BLOCKPRUNER | LAYER | 50% | 47.43 | 58.76 | 36.76 | 42.51 | **28.24** | 42.74 |
| OURS | ALL | | **55.33** | **60.34** | **39.97** | **43.73** | 28.07 | **45.48** |

*Table 7.* Zero-shot performance of the compressed Vicuna-13B. Bold results highlight the best performance.

| VICUNA-13B | PRUNING UNIT | SPARSITY | WINOGRANDE | PIQA | HELLASWAG | ARC-E | ARC-C | AVG |
|---|---|---|---|---|---|---|---|---|
| DENSE | - | 0% | 71.59 | 79.05 | 77.51 | 74.87 | 50.68 | 70.74 |
| SPARSEGPT | 2:4 | 50% | 64.48 | 72.14 | 60.08 | 63.72 | 37.03 | 59.49 |
| WANDA | 2:4 | | 67.17 | 74.10 | 64.07 | 66.20 | 42.66 | 62.84 |
| SLICEGPT | ROW&COLUMN | | 65.51 | 64.20 | 50.51 | 56.94 | 33.62 | 54.15 |
| LLM-PRUNER | ROW&COLUMN | | 59.19 | 77.04 | 65.48 | 63.89 | 40.61 | 61.24 |
| BLOCKPRUNER | LAYER | 30% | 62.43 | **73.67** | **64.90** | 64.94 | **40.78** | 61.34 |
| OURS | ALL | | **65.35** | 73.39 | 63.04 | **69.99** | 40.44 | **62.44** |
| SLICEGPT | ROW&COLUMN | | 58.72 | 58.27 | 38.22 | 44.49 | 27.05 | 45.35 |
| LLM-PRUNER | ROW&COLUMN | | 54.38 | 72.69 | **57.00** | 51.88 | 34.73 | 53.99 |
| BLOCKPRUNER | LAYER | 40% | 58.88 | 67.68 | 53.99 | 54.21 | 34.90 | 53.93 |
| OURS | ALL | | **60.62** | **69.10** | 53.39 | **63.22** | **36.52** | **56.56** |
| SLICEGPT | ROW&COLUMN | | 53.28 | 54.52 | 31.73 | 35.19 | 24.40 | 39.82 |
| LLM-PRUNER | ROW&COLUMN | | 52.09 | **65.07** | 44.59 | 39.06 | 28.75 | 45.91 |
| BLOCKPRUNER | LAYER | 50% | 52.41 | 62.02 | 40.81 | 42.59 | 28.24 | 45.21 |
| OURS | ALL | | **56.12** | 64.04 | **45.10** | **50.08** | **32.25** | **49.52** |

*Table 8.* Zero-shot performance of the compressed LLaMA3-8B. Bold results highlight the best performance.

| LLAMA3-8B | PRUNING UNIT | SPARSITY | WINOGRANDE | PIQA | HELLASWAG | ARC-E | ARC-C | AVG |
|---|---|---|---|---|---|---|---|---|
| DENSE | - | 0% | 72.69 | 80.79 | 79.19 | 77.69 | 53.41 | 72.75 |
| SPARSEGPT | 2:4 | 50% | 64.25 | 68.77 | 55.40 | 59.34 | 32.85 | 56.12 |
| WANDA | 2:4 | | 59.67 | 65.94 | 47.93 | 50.04 | 28.67 | 50.45 |
| SLICEGPT | ROW&COLUMN | | 58.72 | 57.78 | 40.68 | 45.20 | 28.75 | 46.22 |
| LLM-PRUNER | ROW&COLUMN | | 53.67 | 69.15 | 45.06 | 43.31 | 27.65 | 47.76 |
| BLOCKPRUNER | LAYER | 30% | 54.54 | 67.03 | 46.54 | 49.41 | 27.99 | 49.10 |
| SLEB | BLOCK | | **61.01** | 67.74 | **55.53** | 47.10 | 32.25 | 52.72 |
| OURS | ALL | | 60.30 | **69.10** | 54.40 | **54.76** | **33.19** | **54.34** |
| SLICEGPT | ROW&COLUMN | | 52.49 | 53.59 | 33.12 | 35.94 | 22.61 | 39.55 |
| LLM-PRUNER | ROW&COLUMN | | 51.54 | 62.73 | 36.25 | 36.87 | 23.46 | 42.17 |
| BLOCKPRUNER | LAYER | 40% | 50.67 | 58.76 | 36.18 | 36.41 | 24.06 | 41.21 |
| SLEB | BLOCK | | 51.54 | 60.88 | 40.33 | 40.78 | 26.88 | 44.02 |
| OURS | ALL | | **56.27** | **64.20** | **44.74** | **46.51** | **27.30** | **47.80** |
| SLICEGPT | ROW&COLUMN | | 50.04 | 52.23 | 29.18 | 31.31 | 21.67 | 36.88 |
| LLM-PRUNER | ROW&COLUMN | | 51.22 | **60.34** | 31.74 | 33.00 | 24.74 | 40.20 |
| BLOCKPRUNER | LAYER | 50% | 48.62 | 55.22 | 29.12 | 30.01 | 21.59 | 36.91 |
| SLEB | BLOCK | | 50.67 | 55.98 | 31.54 | 31.19 | **24.74** | 38.82 |
| OURS | ALL | | **54.14** | 59.96 | **38.36** | **37.84** | 24.15 | **42.89** |

*Table 9.* Zero-shot performance of the compressed LLaMA3-70B. Bold results highlight the best performance.

| LLAMA3-70B | PRUNING UNIT | SPARSITY | WINOGRANDE | PIQA | HELLASWAG | ARC-E | ARC-C | AVG |
|---|---|---|---|---|---|---|---|---|
| DENSE | - | 0% | 80.35 | 84.55 | 84.88 | 85.86 | 64.33 | 79.94 |
| SPARSEGPT | 2:4 | 50% | 75.37 | 78.35 | 72.30 | 78.37 | 51.37 | 71.15 |
| WANDA | 2:4 | | 71.67 | 79.27 | 73.33 | 76.01 | 49.06 | 69.86 |
| SLICEGPT | ROW&COLUMN | | 69.14 | 65.72 | 55.01 | 64.69 | 42.58 | 59.42 |
| SLEB | BLOCK | | 74.59 | 77.48 | 74.03 | 72.90 | 48.12 | 69.42 |
| BLOCKPRUNER | LAYER | 30% | 71.90 | 76.12 | 73.81 | 69.40 | 47.44 | 67.73 |
| OURS | ALL | | **75.53** | **80.69** | **80.38** | **77.78** | **52.05** | **73.28** |
| SLICEGPT | ROW&COLUMN | | 61.09 | 58.49 | 43.93 | 47.85 | 32.85 | 48.81 |
| SLEB | BLOCK | | 68.75 | 71.93 | 63.49 | 60.65 | 37.63 | 60.48 |
| BLOCKPRUNER | LAYER | 40% | 66.38 | 71.06 | 64.76 | 59.60 | 37.80 | 59.91 |
| OURS | ALL | | **72.93** | **77.26** | **73.27** | **73.23** | **47.44** | **68.82** |
| SLICEGPT | ROW&COLUMN | | 54.70 | 54.95 | 33.11 | 36.74 | 25.77 | 41.05 |
| SLEB | BLOCK | | 56.20 | 66.92 | 49.99 | 48.32 | 29.27 | 50.13 |
| BLOCKPRUNER | LAYER | 50% | 55.49 | 64.85 | 51.09 | 45.45 | 30.29 | 49.43 |
| OURS | ALL | | **68.82** | **72.63** | **61.72** | **61.78** | **39.33** | **60.85** |

## C. Ablation Studies

### C.1. Ablation Study on Criteria for Quantifying Module Interactions

We use the Frobenius norm (He et al., 2017) instead of transfer entropy as the new criterion to measure the change of LLM hidden state. We conduct evaluations on five different zero-shot datasets. As shown in Table 10, TE consistently preserves better model performance compared to the Frobenius norm.

*Table 10.* Different criteria for quantifying block interactions. Bold results highlight the best performance.

| LLaMA2-7B | Pruning Unit | Sparsity | Winogrande | PIQA | Hellaswag | Arc-e | Arc-c | Avg |
|---|---|---|---|---|---|---|---|---|
| Frobenius norm | All | 50% | **54.54** | 61.75 | 39.29 | 42.05 | 26.28 | 44.78 |
| TE(ours) | All | | 53.83 | **61.81** | **40.46** | **43.94** | **27.56** | **45.52** |

### C.2. Ablation Study on Search Method

We compare Depth-First Search (DFS) with an intuitive solution - greedy search. We observe greedy search tends to prioritize removing entire blocks during the compensation phase to quickly satisfy the remaining pruning ratio in Equation (5). However, due to the lack of a backtracking mechanism, greedy search selects the seemingly optimal option at each step but is prone to getting stuck in local optima. Our ablation study in Table 11 further supports this observation, demonstrating that DFS consistently outperforms greedy search in terms of pruning effectiveness.

*Table 11.* Different search methods. Bold results highlight the best performance.

| LLaMA2-7B | Pruning Unit | Sparsity | Winogrande | PIQA | Hellaswag | Arc-e | Arc-c | Avg |
|---|---|---|---|---|---|---|---|---|
| Greedy Search | All | 50% | 53.67 | 61.53 | 39.96 | 41.75 | 27.22 | 44.82 |
| DFS | All | | **53.83** | **61.81** | **40.46** | **43.94** | **27.56** | **45.52** |

### C.3. Ablation Study on Hierarchical Pruning

In this ablation study, we adopt the Hybrid pruning strategy proposed in Sec. 3.2 to jointly prune both blocks and layers, thereby implementing hierarchical pruning. However, for the remaining blocks, we implement the approach described in Sec. 3.3, pruning only the rows&columns of the weight matrices without applying dynamic ratio allocation. Instead, a uniform pruning ratio is used across all of them. To compare with pruning single structure unit, without using hierarchical pruning, we apply a uniform pruning ratio across all blocks and prune only the entire rows&columns of the weight matrices. The corresponding results are shown in Table 12.

*Table 12.* Zero-shot performance of the Hierarchical Pruning. Bold results highlight the best performance.

| LLaMA2-7B | Pruning Unit | Sparsity | Winogrande | PIQA | Hellaswag | Arc-e | Arc-c | Avg |
|---|---|---|---|---|---|---|---|---|
| Single unit | Row&Column | 50% | 51.07 | 55.60 | 29.60 | 33.67 | 24.15 | 38.81 |
| Multiple units | All | | **55.09** | **61.32** | **40.50** | **43.22** | **27.13** | **45.45** |

### C.4. Ablation Study on Dynamic Ratio

To validate the effectiveness of dynamic pruning ratios, we build upon the Hierarchical Pruning setup from Appendix C.3 by incorporating dynamic ratio assignment. The corresponding ablation results are presented in Table 13.

*Table 13.* Zero-shot performance of dynamic ratios. Bold results highlight the best performance.

| LLaMA2-7B | Pruning Unit | Sparsity | Winogrande | PIQA | Hellaswag | Arc-e | Arc-c | Avg |
|---|---|---|---|---|---|---|---|---|
| Static ratio | All | 50% | **55.09** | 61.32 | **40.50** | 43.22 | 27.13 | 45.45 |
| Dynamic ratio | All | | 53.83 | **61.81** | 40.46 | **43.94** | **27.56** | **45.52** |

