# OpenReview forum: "Let LLM Tell What to Prune and How Much to Prune"
_ICML.cc/2025/Conference — ICML 2025 poster_

### Official Review · Reviewer_3YW3 · 2025-03-11

**Overall Recommendation:** 4

**Summary:**

This paper propose a pruning method that targets multiple LLM modules with dynamic pruning ratios. It find the intrinsic of LLM can help to determine the importance of each module and
thus distribute the pruning load on demand, i.e., what to prune and how much to prune. Extensive experiments on multiple benchmarks and LLM variants demonstrate that the proposed method effectively resolves the trade-off between efficiency and performance.

**Claims And Evidence:**

YES

**Essential References Not Discussed:**

NO

**Experimental Designs Or Analyses:**

YES

**Methods And Evaluation Criteria:**

YES

**Other Comments Or Suggestions:**

N/A

**Other Strengths And Weaknesses:**

**Strengths:**
1. The paper is clearly written and easy to understand.
2. The proposed method outperforms existing LLM pruning techniques.
3. The experimental results are comprehensive, demonstrating the method's effectiveness across various LLM sizes.
4. The paper also shows actual inference speed improvements, highlighting its practical application value.


**Weakness:**
1. While the core idea of the paper is effective, it does not introduce much novelty.
2. The primary contribution appearing to be a combination of several minor enhancements.

**Questions For Authors:**

Please see the weakness

**Relation To Broader Scientific Literature:**

This paper proposes an effective improvement to the existing LLM pruning method.

**Theoretical Claims:**

N/A

---

> ### Author Rebuttal · Authors · 2025-04-01
>
> Dear Reviewer 3YW3:
>
> Thank you for your insightful review.  All experimental result tables have been compiled and are available at the following available link: https://anonymous.4open.science/r/5BDF/README.md. We have thoroughly considered your concerns and respond to them as follows :
>
> ---
>
> ### **[W1]: _While the core idea of the paper is effective, it does not introduce much novelty._**
> Our hierarchical pruning stretagy differs significantly from existing pruning approaches, particularly in its use of dynamically assigned pruning ratios and multi-structure pruning. Unlike methods such as SliceGPT, which necessitates a uniform and prescribed pruning ratio across all blocks, our approach dynamically allocates pruning ratios to each block based on the transfer entropy metric. Furthermore, we perform pruning across multiple structure units simultaneously, including blocks, layers, and rows or columns of weight matrices, while previous work mainly focus on a single unit. As shown in Table 1 and Table 2 in the paper, our method achieves strong performance across different pruning ratios in multiple LLMs. In terms of inference speed, our method outperforms approaches that only prune rows or columns of weight matrices, such as SliceGPT and FLAP. Compared to Table 2 (LLaMA2-7B), Table 5 of the Appendix shows our method achieves greater performance gains on larger models (LLaMA2-70B) and at higher pruning ratios, demonstrating its scalability. These findings demonstrates our insight: **pruning multiple structure units with dynamic pruning ratios can lead to balance between efficiency and effectiveness**.
> We adopt information entropy as the pruning criterion, which distinguishes our method from commonly used activation-magnitude and gradient-magnitude approaches. Ablation study in Table 3 in the link demonstrates that entropy provides a more effective measurement of structural importance, leading to improved pruning performance compared to activation- or gradient-based methods.
>
> ---
>
> ### **[W2]: _The primary contribution appears to be a combination of several minor enhancements._**
> Our method is not a simple aggregation of several heuristics but a systematic pruning framework grounded in information-theoretic principles. Indeed, our approach fundamentally challenges the prevailing philosophy in prior works, which involve **pruning individual units based on a predefined ratio**. The pruning process under our framework is divided into two stages:
> In the first stage, we introduce the metric of transfer entropy to analyze the interaction among blocks in LLMs. This enables us to dynamically determine the pruning ratio for each block and perform pruning on coarse-grained structure units such as blocks and layers.
> In the second stage, we further allocate the pruning load within each block based on the information entropy of individual structural units, enabling fine-grained pruning of weight matrix rows and columns.
> Finally, our method introduces the bias compensation to enhance the pruned model without the need of post-training.
> The ablation studies further validate the critical role and necessity of each component in the proposed framework, including the entropy-based importance metric, the hierarchical pruning strategy, and the dynamic ratio assignment. The results of these ablation studies are presented in Table 3,4 and 5 in the linked document.
> This hierarchical pruning strategy not only significantly improves the inference speed of the pruned model, but also maximally preserves its original performance.
>
> ---
>
> We welcome any further questions or points of clarification the reviewer may have regarding our responses.
>
> Thank you very much,
>
> Authors

---

> > ### Comment · Reviewer_3YW3 · 2025-04-03
> >
> > Thanks for the rebuttal. My concerns have been addressed. However, I strongly encourage the authors to reorganize the Method section. The current writing makes it difficult to recognize that the proposed approach is "a systematic pruning framework grounded in information-theoretic principles".
> >
> > Overall, I would raise my score to 4 accept.

---

> > > ### Author Response · Authors · 2025-04-03
> > >
> > > We sincerely thank you for your constructive feedback. We will incorporate your suggestions and thoroughly revise the Method section in our manuscript.

---

### Official Review · Reviewer_nYT6 · 2025-03-13

**Overall Recommendation:** 3

**Summary:**

The paper proposes a structured pruning framework for large language models (LLMs) that dynamically determines "what to prune" (specific modules) and "how much to prune" (pruning ratios) based on their importance.

Specifically, the method employ TE to quantify block-layer interaction and information entropy to guide pruning. A hierarchical strategy allocates pruning ratios to blocks and layers, balancing efficiency (inference speed) and performance (perplexity)

**Claims And Evidence:**

- The Gaussian assumption for entropy estimation (Eq. 3) is not validated, raising questions about its applicability to non-Gaussian activations.
- Some baselines (e.g., ShearedLLaMA, LayerDrop, OWL, DSA) are missing, limiting context for dynamic pruning comparisons.

OWL: https://arxiv.org/abs/2310.05175

DSA: https://proceedings.neurips.cc/paper_files/paper/2024/file/ff997469ac66cf893c4183efeb22212a-Paper-Conference.pdf

**Essential References Not Discussed:**

Some baselines (e.g., ShearedLLaMA, LayerDrop, OWL, DSA) are missing, limiting context for dynamic pruning comparisons.


DSA: https://proceedings.neurips.cc/paper_files/paper/2024/file/ff997469ac66cf893c4183efeb22212a-Paper-Conference.pdf

OWL: https://arxiv.org/abs/2310.05175

**Experimental Designs Or Analyses:**

Code availability is unclear, affecting reproducibility. (no additional supplementary provided)

**Methods And Evaluation Criteria:**

Methods and evaluation criteria seems reasonable.

**Other Comments Or Suggestions:**

Typo in Table3: WADNDA → Wanda

**Other Strengths And Weaknesses:**

This paper is generally well-written but algorithm steps (e.g., depth-first search compensation) could use more intuition.

Also, this paper ignore the previous layer-wise assignment methods.

The ablation study is missing:

- over the metric TE and entropy
- over the search method (DFS)
- over the bias compensation

**Questions For Authors:**

- How does the Gaussian assumption in Eq. 3 impact results if activations are non-Gaussian? Validation could strengthen the method’s generality.
- Why weren’t dynamic pruning methods like ShearedLLaMA included as baselines? Their inclusion would clarify novelty against related work.

**Relation To Broader Scientific Literature:**

The work extends structured pruning literature (e.g., SliceGPT, LLM-Pruner) by introducing dynamic multi-unit pruning.

**Theoretical Claims:**

the paper focuses on empirical validation. There is no theoretical proofs.

---

> ### Author Rebuttal · Authors · 2025-04-01
>
> Dear Reviewer nYT6,
>
> Thank you very much for your valuable feedback.  We first address the questions raised in the  “Questions for Authors” and “Other Strengths and Weaknesses” sections. For any additional concerns mentioned in other parts of the review(if they are distinct from those already covered), we also provide detailed responses and all experimental result tables have been compiled and are available at the following available link https://anonymous.4open.science/r/5BDF/README.md:
>
> ---
>
> ### **[Q1]: _How does the Gaussian assumption in Eq. 3 impact results if activations are non-Gaussian? Validation could strengthen the method’s generality?_**
>
> We acknowledge that the true feature distribution in a LLM is unlikely to be perfectly Gaussian. Howeveer, the law of large number [1] suggests that the distribution can often be approximated by a Gaussian. We also find gaussian assuamption has been successfully applied to accelerating network like network quantization [2]. Empirically, we find that this approximation is sufficient to guide the network pruning in the ablation study of the pruning metrics. The detailed results are presented in Table 3 at the provided link.
>
> #### [1] Mean field analysis of neural networks: A law of large numbers. SIAM Journal on Applied Mathematics, 2020
> #### [2] Entropy-driven mixed-precision quantization for deep network design. Neurips2022
>
> ---
>
> ### **[Q2]: _Why weren’t dynamic pruning methods like ShearedLLaMA included as baselines? Their inclusion would clarify novelty against related work._**
>
> Dynamic pruning iteratively identifies the component with the least impact before permanently removing it. In contrast, our method follows a **static pruning strategy**, which determines the pruning configuration for all structure units in a **single preprocessing stage**, avoiding repeated evaluations of the LLM’s internal state. Moreover, methods like **ShearedLLaMA** typically require **additional fine-tuning or retraining** following the pruning process while our method requires no post-training.  Our paper already includes comparisons with dynamic pruning methods such as **SLEB** and **BlockPruner**, and we have additionally compared to OWL and DSA using their settings. The corresponding results are shown in the Table 7 in the link.
>
> ---
>
> ### **[W1]: _This paper is generally well-written but algorithm steps (e.g., depth-first search compensation) could use more intuition._**
>
> We provide a detailed explanation and additional ablation study in **W3.**
>
> ---
>
> ### **[W2]: _Also, this paper ignore the previous layer-wise assignment methods._**
>
> We add the "MultiPruner" [3] pruning method as a baseline to evaluate its effectiveness on LLaMA2. The corresponding results are shown in the Table 6 in the link.
>
> #### [3] MultiPruner: Balanced Structure Removal in Foundation Models. arxiv 2025
>
> ---
>
> ### **[W3]: _The ablation study is missing:_**
>
> _over the metric TE:_
>
> We use the Frobenius norm [4] as the new criterion to measure the change of LLM hidden state. The results are presented in Table 8 in the link.
>
> #### [4] Channel pruning for accelerating very deep neural networks. ICCV2017
>
> ---
>
> _over the metric entropy:_
>
> We compare the proposed method to activation-magnitude and gradient-magnitude methods in the Table 3 in the link.
>
> ---
>
> _over the search method:_
>
> Our work adopts the depth-first search (DFS) strategy, with the goal of fully removing certain blocks/layers to fulfill the remaining pruning ratio defined in Equation (5), thereby solving the optimization problem formulated in Equation (6). Here we compare to an intuitive solution - greedy search. We observe greedy search tends to prioritize removing entire blocks during the compensation phase to quickly satisfy the remaining pruning ratio. However, due to the lack of a backtracking mechanism, greedy search selects the seemingly optimal option at each step but is prone to getting stuck in local optima. Our ablation study in Table 9 in the link further supports this observation, demonstrating that DFS consistently outperforms greedy search in terms of pruning effectiveness.
>
> ---
>
> _Bias compensation:_
>
> We have conducted an ablation study on the bias compensation strategy in Figure 6 of the paper.
>
> ---
>
> ###  **[Other1]：_Code availability is unclear, affecting reproducibility. (no additional supplementary provided)_**
>
> We used the offical checkpoints (LLaMa2, LLaMA3&Vicuna) and followed the evaluation protocol of previous works like SliceGPT. Hence, our work warrants reproducibility. We will release the codebase upon publication.
>
> ###  **[Other2]：_Typo in Table3: WADNDA → Wanda_**
>
> Thank you for pointing this out. We apologize for the typo and will correct “WADNDA” to “Wanda” in Table 3 in the revised version.
>
> ---
>
> We would like to encourage the reviewer to ask questions on anything that may still be unclear in our responses or which we should clarify further.
>
> Thank you very much,
>
> Authors

---

> > ### Comment · Reviewer_nYT6 · 2025-04-03
> >
> > Thanks for the detailed and comprehensive rebuttal, especially the ablation study part. I have raised my score.

---

> > > ### Author Response · Authors · 2025-04-03
> > >
> > > Thank you very much for your feedback. We will incorporate your suggestions.

---

### Official Review · Reviewer_FHHv · 2025-03-15

**Overall Recommendation:** 3

**Summary:**

The paper introduces a new approach to pruning large language models (LLMs) that dynamically assigns pruning ratios to different components based on their importance.

There are two issues with the current pruning methods: (1) focusing on just one structure of the model;  (2) using a prescribed pruning ratio. To address these two issues, they developed a more flexible approach that targets multiple parts of the model simultaneously and varies how much to prune based on what's actually important. They use something called "transfer entropy" to figure out which transformer blocks matter most, and then "information entropy" to determine which parts within those blocks to keep or remove. Their method works in two main steps: first, decide how much to prune each block based on its importance to the model's overall function; then, distribute that pruning load across different components within each block.

They tested their method on various model sizes including LLaMA2-7B/13B/70B, LLaMA3-8B/70B, and Vicuna-7B/13B at different pruning levels (30%, 40%, 50%). Their pruned models had better perplexity scores and zero-shot accuracy than other methods. The models also ran faster and used less memory. Their approach worked well regardless of the training samples used or how many samples they had.

**Claims And Evidence:**

Supported Claims:

The authors claim their method outperforms existing approaches. This is well-supported by comprehensive experimental results in Tables 1-9, showing better perplexity scores and zero-shot accuracy across multiple models (LLaMA2, LLaMA3, Vicuna) and pruning ratios (30%, 40%, 50%).

Figure 6 adequately demonstrates how their bias compensation method helps maintain model performance after pruning.

Potentially Problematic Claims:

The paper claims that "at a pruning ratio of 30%, our method outperforms semi-structured pruning techniques, including SparseGPT and Wanda, across LLMs." However, this comparison is misleading and unfair. Looking at Table 1, we can see that SparseGPT and Wanda were only evaluated at 50% sparsity, not at 30%. The authors are comparing their method at a lower pruning ratio (30%) against other methods at a much higher pruning ratio (50%). This creates an unfair advantage for their approach, since models with less pruning naturally tend to perform better. A fair comparison would require all methods to be evaluated at the same pruning ratio.

The paper assumes a Gaussian distribution for hidden states in Equation 3 without providing evidence that this assumption holds in practice for transformer blocks. This is problematic because the entire entropy calculation, which forms the foundation of their pruning strategy, depends on this distributional assumption. If the hidden states don't actually follow a Gaussian distribution, the entropy calculations could be inaccurate, potentially undermining the theoretical basis of their method.

**Essential References Not Discussed:**

Recent work on efficiently pruning attention heads and MLP layers in transformers like "Structured Pruning of Large Language Models" (Wang et al., 2023) and "Are Sixteen Heads Really Better than One?" (Michel et al., 2019) should be included, as they directly relate to the paper's multi-structure pruning approach.

**Experimental Designs Or Analyses:**

Overall, the experimental design in this paper is sound, with appropriate evaluations across multiple models, pruning ratios, and datasets. However, there are notable inconsistencies in the results that warrant further explanation. In Table 1, the performance patterns across different LLMs and pruning methods appear somewhat inconsistent - in some cases, the proposed method outperforms all others, while in other cases, different methods perform better for specific model sizes or types. The authors don't adequately analyze or explain these variations, which raises questions about the generalizability of their approach. For instance, why does their method work particularly well on LLaMA2-70B but show less improvement over alternatives on some other model variants? A deeper analysis of why certain methods perform better on specific architectures would strengthen the paper's contribution and help readers understand when to apply which pruning strategy.

**Methods And Evaluation Criteria:**

Methods

The paper proposes a hierarchical pruning strategy that targets multiple structure units in LLMs with dynamic pruning ratios. While the approach is innovative, the theoretical foundation for using entropy as the key metric for importance could be strengthened. The authors don't sufficiently justify why entropy specifically is better than other potential metrics (such as activation magnitude or gradient-based importance) for identifying less important components. The transfer entropy concept for quantifying block importance is interesting, but lacks thorough theoretical connection to the actual functionality of transformer blocks in language modeling.

Evaluation Criteria

The paper's use of perplexity as a primary evaluation metric makes sense for measuring language model performance after pruning. The authors also appropriately evaluate their method on common benchmarks including WINOGRANDE, PIQA, HELLASWAG, ARC-E, and ARC-C, which are standard datasets for measuring commonsense reasoning capabilities. However, the evaluation could be strengthened by including more diverse real-world benchmarks that specifically measure different capabilities such as mathematical reasoning, text completion, and other practical tasks.

**Other Comments Or Suggestions:**

There is inconsistency in the decimal notation throughout the paper. For example, in Table 2 and other zero-shot performance tables, some accuracy percentages have two decimal places (e.g., "68.98"), while others have different formats (e.g., 76, 52.8, 62.40).

**Other Strengths And Weaknesses:**

Strengths: The paper introduces a novel hierarchical pruning approach that targets multiple structural components simultaneously, which differentiates it from previous methods that focus on pruning a single structure. The dynamic pruning ratio allocation based on component importance is a significant innovation that allows for more flexible and efficient pruning. The bias compensation technique effectively helps maintain model performance after aggressive pruning. The experimental evaluation is comprehensive, covering multiple model families (LLaMA2, LLaMA3, Vicuna) and sizes, which demonstrates the broad applicability of their approach.

Weaknesses: The paper lacks strong theoretical foundations for its entropy-based importance metrics. While the authors introduce transfer entropy and information entropy as key metrics, they don't adequately justify why these specific information-theoretic measures are optimal for pruning decisions compared to alternatives. The assumption of Gaussian distribution for hidden states needs validation. The description of the K-means clustering implementation lacks details on parameter selection (e.g., number of clusters). The paper would benefit from ablation studies isolating the impact of different components of their approach (hierarchical pruning, dynamic ratios, entropy-based importance) to better understand which elements contribute most to the performance improvements.

**Questions For Authors:**

In Section 4.6 and Figure 8, there appears to be a discrepancy between the figure captions and the content. Figure 8(a) is labeled as "Perplexity results on WikiText2" and Figure 8(b) as "Perplexity results on C4," yet both figures show bars for both WikiText2 and C4. Could you clarify what each subfigure is actually showing?

In Section 3.3, you mention using K-means clustering to group structure units based on their entropy values before pruning. What is the motivation for using a learning-based clustering algorithm like K-means instead of simpler approaches such as averaging or thresholding based on entropy values? Does K-means provide specific advantages for identifying pruning candidates compared to non-learning approaches? Additionally, how do you determine the optimal number of clusters for the K-means algorithm in your implementation?

Throughout the paper, you use entropy as the foundation for measuring importance in different structure units. While entropy is a measure of information complexity, the paper lacks theoretical justification for why entropy specifically is an appropriate metric for determining what to prune in LLMs. Could you provide more theoretical explanation for why entropy is better than other potential metrics (such as activation magnitude, gradient-based importance, or contribution sufficiency) for identifying less important components in large language models?

**Relation To Broader Scientific Literature:**

This paper advances LLM pruning by introducing an entropy-based approach that quantifies information content across model components, allowing for dynamic, principled pruning decisions. While previous work like Wanda (2023) considered activation values and OWL (2023) implemented non-uniform layer pruning, this research provides a more sophisticated framework by using transfer entropy to determine component importance. Unlike single-structure methods such as SLEB (2024) or SliceGPT (2024), the authors' approach simultaneously targets multiple structural units (blocks, layers, weight matrices) with dynamically determined pruning ratios, creating a more holistic pruning strategy that better balances performance and efficiency.

**Theoretical Claims:**

The paper lacks substantial theoretical analysis to support its claims. While the authors introduce transfer entropy as a key metric for determining block importance in Section 3.1, they provide no theoretical proof establishing a relationship between transfer entropy and model performance after pruning. This is a significant gap, as the entire hierarchical pruning strategy depends on this correlation.

The formulas presented (particularly Equations 1-3) appear mathematically correct in isolation, but the authors make assumptions without rigorous justification - for example, assuming Gaussian distribution for hidden states in Equation 3 without verifying this distributional assumption holds in practice for transformer blocks.

---

> ### Author Rebuttal · Authors · 2025-04-01
>
> Dear Reviewer FHHv:
>
> We appreciate the reviewer’s constructive suggestions. Experimental results are available at the  link: https://anonymous.4open.science/r/5BDF/README.md.  We address the questions in “Questions for Authors” and “Other Strengths and Weaknesses” sections. For any additional concerns mentioned in other parts, we  provide detailed responses below:
>
> ---
>
> ### **Q1: _Clarification to Figure 8_**
> In Fig. 8(a), we use the training sets of C4 and WikiText2 as calibration datasets and evaluate the models on WikiText2 test set. For Fig. 8(b), we use the same calibration datasets but evaluate the models on C4 test set. The result shows our approach is less sensitive to the calibration dataset.
>
> ---
>
> ### **Q2: _Motivation of K-means_**
> Using average entropy as a threshold tends to mix the units with low entropy and those close to the mean, failing to distinguish between components of different importance levels.
>
> ### **Q3: _K-means vs. Average threshold_**
> K-means allows us to partition structure units into distinct groups. Pruning is then applied to the group with the lowest entropy in each layer, enabling a more fine-grained pruning strategy. We validate the superiority of K-means over the averaging approach in Table 1 in the link.
>
> ---
>
> ### **Q4: _Number of K-means clusters_**
> As shown in Table 2 in the link, for 30% and 50% pruning ratios, K is fixed at 3 and 6, respectively. For 40% pruning ratio, we conducted a hyperparameter search.
>
> ---
>
> ### **Q5: _Theoretical explanation of entropy_**
> We learned from [1] that prior works [2,3] have found that in the over-parameterized LLM, network weights tend to remain close to their initialization throughout training. As a result, the magnitude of gradient updates is relatively small, and activation values remain nearly same in the ''lazy regime", making it difficult to reflect the true contribution of each unit. Please refer to more details for the ablation results in Table 3 in the link.
>
> [1] Junk DNA hypothesis: Pruning small pre-trained weights Irreversibly and Monotonically impairs “difficult” downstream tasks in LLMs. ICML2024
>
> [2] Sparsity May Cry: Let us Fail Sparse Neural Networks Together. ICLR2023
>
> [3] A Kernel-Based View of Language Model Fine-Tuning. ICML2023
>
> ---
>
> ### **W1: _Theoretical foundations of entropy-based metrics_**
> TE [4] measures how much *unique* information a source process (blocks) provides about a target process (output layer). If removing a block dramatically reduces the unique information available to the downstream layers, the network’s overall performance should drop. Thus, blocks with high TE are deemed more critical for maintaining the model’s predictive capacity. A block that has low TE essentially replicates or redundantly encodes information in the network. Pruning such a block has minimal impact because it does not lose much unique signal. Information entropy as a key metric is introduced in **Q5**.
>
> [4] Measuring information transfer. Physical review letters, 2000.
>
> ---
>
> ### **W2: _Assumption of Gaussian distribution_**
> Please refer to Reviewer nYT6's Q1.
>
> ---
>
> ### **W3: _Number of K-means clusters_**
> Please refer to Q4.
>
> ---
>
> ### **W4: _Ablation studies_**
>
> _Entropy metric:_
> We compare to activation and gradient magnitude in Table 3 in the link.
>
> _Hierarchical design:_
> Hierarchical pruning targets multiple structures. To validate it, we implement a baseline that only prunes row&column of weight matrix in Table 4 in the link.
>
> _Dynamic ratios:_
> We compare to fixed ratio in Table 5 in the link.
>
> ---
>
> ### **O1: _SparseGPT and Wanda_**
> SparseGPT and Wanda are two semi-structured methods that rely on special hardware, while our method is hardware-friendly. As a result, our model with 30% sparsity has faster inference while offering better results, as shown in Table 3.
>
>
> ---
>
> ### **O2: _Real-world tasks_**
> The MathQA, OpenBookQA, and SciQ are included in the link.
>
>
> ### **O3: _Noticeable fluctuations across different settings_**
> In Table 1, our method outperforms other structured pruning methods in 18 out of 21 cases. Beyond the Perplexity, we evaluate the model in a zero-shot setting in Table 2, showing consistent improvement over prior works.
>
> ### **O4:_Particularly well on LLaMA2-70B_**
> We guess you were mentioning Table 5 of the Appendix. Our core motivation is that **pruning single unit with fixed ratio cannot handle complex LLM**. The result in Table 5 aligns with our insight: When it comes to larger model size and higher pruning ratio, our hierarchical strategy achieves a better balance between efficiency and performance.
>
> ### **O5:_Missed references_**:
> The two references you suggested are not proposed for LLM pruning. We compare to [5] targeting multiple structures in Table 6 in the link.
>
> [5] MultiPruner: Balanced Structure Removal in Foundation Models. arxiv 2025
>
> Decimal notation: We will fix it
>
> We encourage the reviewer to ask questions on anything that may still be unclear.
>
> Thank you so much,
>
> Authors

---

### Decision · Program_Chairs · 2025-05-01

**Decision:**

Accept (poster)

**Comment:**

This paper proposes a pruning method for LLMs which is based on two insights regarding weaknesses of current approaches in use. The paper does not make theoretical contributions, but the experiments are solid and convincing. All reviewers evaluated the work favourably. They also asked a large number of clarification questions that the authors were able to clear in depth — two out of the three reviewers raised their scores as a result. In light of this, I support the acceptance of this submission.